# Distinct Genomic Expression Signatures after Low-Force Electrically Induced Exercises in Persons with Spinal Cord Injury

**DOI:** 10.3390/ijms251810189

**Published:** 2024-09-23

**Authors:** Michael A. Petrie, Manish Suneja, Richard K. Shields

**Affiliations:** 1Department of Physical Therapy and Rehabilitation Science, Carver College of Medicine, The University of Iowa, Iowa City, IA 52242, USA; michael-petrie@uiowa.edu; 2Department of Internal Medicine, Carver College of Medicine, The University of Iowa, Iowa City, IA 52242, USA; manish-suneja@uiowa.edu

**Keywords:** rehabilitation, low force exercise, gene expression, methylation expression

## Abstract

People with a spinal cord injury are at an increased risk of metabolic dysfunction due to skeletal muscle atrophy and the transition of paralyzed muscle to a glycolytic, insulin-resistant phenotype. Providing doses of exercise through electrical muscle stimulation may provide a therapeutic intervention to help restore metabolic function for people with a spinal cord injury, but high-frequency and high-force electrically induced muscle contractions increase fracture risk for the underlying osteoporotic skeletal system. Therefore, we investigated the acute molecular responses after a session of either a 3 Hz or 1 Hz electrically induced exercise program. Ten people with a complete spinal cord injury completed a 1 h (3 Hz) or 3 h (1 Hz) unilateral electrically induced exercise session prior to a skeletal muscle biopsy of the vastus lateralis. The number of pulses was held constant. Tissue samples were analyzed for genomic and epigenomic expression profiles. There was a strong acute response after the 3 Hz exercise leading to the upregulation of early response genes (NR4A3, PGC-1α, ABRA, IRS2, EGR1, ANKRD1, and MYC), which have prominent roles in regulating molecular pathways that control mitochondrial biogenesis, contractile protein synthesis, and metabolism. Additionally, these genes, and others, contributed to the enrichment of pathways associated with signal transduction, cellular response to stimuli, gene expression, and metabolism. While there were similar trends observed after the 1 Hz exercise, the magnitude of gene expression changes did not reach our significance thresholds, despite a constant number of stimuli delivered. There were also no robust acute changes in muscle methylation after either form of exercise. Taken together, this study supports that a dose of low-force electrically induced exercise for 1 h using a 3 Hz stimulation frequency is suitable to trigger an acute genomic response in people with chronic paralysis, consistent with an expression signature thought to improve the metabolic and contractile phenotype of paralyzed muscle, if performed on a regular basis.

## 1. Introduction

People aging with a spinal cord injury (SCI) are at a high risk of developing cardiometabolic non-communicable diseases (NCD), such as obesity, type 2 diabetes mellitus, and metabolic syndrome, which contribute to decreased life expectancy, lower overall life quality, and increased health-care costs [1,2,3,4]. The lack of routine exercise contributes to metabolic dysfunction after a SCI [5,6,7]. Insulin signaling resistance, a hallmark of metabolic dysfunction, results in part from the atrophy and glycolytic phenotype transition that occurs after paralysis because of the loss in routine skeletal muscle activity and exercise leading to a sedentary lifestyle [8,9,10,11,12]. We know there is a benefit to interrupting prolonged periods of sedentary behaviors with bouts of physical activity and exercise, even at low intensities, for people with and without SCI [13,14,15]. However, unlike people without SCI, people with SCI are unable to volitionally control their paralyzed extremities, which creates an additional barrier to engaging the paralyzed muscles in bouts of exercise. This combined with other common barriers, such as (1) increased injury risk from musculoskeletal deterioration [16,17], (2) lack of education, support, or motivation [18,19,20], and (3) limited or no access to the specialized equipment needed to exercise the non-paralyzed or paralyzed extremities, such as expensive functional electrical stimulation cycling or rowing ergometers [20,21], prohibits many people from engaging in bouts of physical activity and exercise. Therefore, there is a need for the development of exercise prescriptions, grounded within exercise physiology principles, that are economical and feasible for deployment in the home.

Electrically induced exercise, a rehabilitation therapy, delivers electrical stimulus pulses to evoke action potentials in motor neurons, causing skeletal muscle contractions. These evoked contractions utilize the existing metabolic and contractile components of skeletal muscle, mimicking the effect of exercise for people with paralysis. Notably, electrically induced exercise activates the insulin-independent pathways critical for the homeostatic balance of blood glucose [11,22,23]; and provides a physiologic challenge to the paralyzed muscle [24,25,26,27], systemic metabolism [13,28], autonomic nervous system [29,30], and cardiovascular system [30,31]. Therefore, electrically induced exercise may offer a key rehabilitation prescription to help maintain health for those living with SCI.

Comparing the acute molecular signatures of electrically induced exercise provides valuable insight into determining the dose required to adequately stress paralyzed skeletal muscle. Skeletal muscle phenotype and oxidative capacity are controlled by genomic and epigenomic regulation, such the expression of peroxisome proliferator-activated receptor gamma coactivator 1 alpha (PGC1α) [32,33], nuclear receptor subfamily 4 group A member 3 (NR4A3) [34,35,36], actin binding rho activating protein (ABRA) [37,38], and insulin receptor substrate 2 (IRS2) [39,40,41]; or the repression of myostatin (MSTN) [42,43]. Electrically induced exercise alters the expression of these important regulatory genes [25,26,44,45]. Further, long-term electrically induced exercise training reduced skeletal muscle atrophy and preserved the oxidative phenotype of acutely paralyzed skeletal muscle in people with SCI [25,46,47,48]. However, the impact that electrically induced exercise can have on chronically paralyzed muscle after the onset of significant musculoskeletal deterioration remains unknown. Unfortunately, people living with SCI have compromised skeletal systems with significant bone loss (osteoporosis) where the high force contractions could lead to debilitating bone fractures [16,49,50]. Therefore, caution is required when developing exercise therapies to minimize the risk of bone fractures while still providing the dose of muscle activity needed to reverse the fatigable, glycolytic phenotype transitions subsequent to prolonged inactivity associated with paralysis [9,24,47,50].

Low-force electrically evoked contractions may be one method to mitigate the risks of applying high forces within a weakened musculoskeletal system and still obtain physiological benefits. This form of electrically induced exercise spaces the stimulus pulses over a longer duration, preventing force summation between consecutive stimulus pulses. It has previously been shown that muscle activity spread over a longer duration can increase energy expenditure and improve systemic glucose regulation compared to short-duration exercises [28,51]. Therefore, a long-duration, low-force electrically induced exercise using stimulation frequencies below 5 Hz [26,52] may provide a safe therapeutic dose of exercise for people with chronic SCI while still providing an adequate physiologic challenge to the paralyzed skeletal muscles [27] and blood glucose regulation [13]. Both 1 Hz and 3 Hz stimulations meet the requirement for an unfused tetany causing a similar level of peak force. However, when keeping the number of pulses constant, 1 Hz stimulation takes 3 h, while 3 Hz stimulation takes only one hour, a meaningful difference as we seek participant compliance with a dose of exercise. We have already established that the 3 Hz protocol is “more acceptable” than the 1 Hz protocol to people with SCI [27], but we have not compared these protocols at the genomic and epigenomic levels in people with chronic paralysis.

Accordingly, we quantified the acute genomic and epigenomic responses 3 h after a 1 or 3 h long low-force electrically induced exercise using a 3 Hz and 1 Hz stimulation frequency (the number of pulses remained constant). Both exercises have been shown to fatigue paralyzed skeletal muscle after just 1000 pulses; however, the extent of fatigue was significantly greater after the 3 Hz protocol as compared to the 1 Hz protocol [27]. Therefore, we hypothesized that the 3 Hz protocol delivered over a 1 h period would provide a stronger acute genomic and epigenomic response in chronic paralyzed skeletal muscle as compared to the 1 Hz protocol delivered over 3 h. It is worth noting that the number of stimuli was held constant between the two protocols (12,000 pulses).

## 2. Results

### 2.1. Genomic Expression Signature

The microarrays were summarized and annotated to 22,870 genes. Three hours after the completion of the 3 Hz exercise, there were 258 genes with an increased differential expression and 274 genes with a decreased differential expression. However, no genes were found to be differentially expressed 3 h after the 1 Hz exercise. Therefore, we used an unadjusted *p*-value with a threshold of 0.004 for the 1 Hz exercise to make general comparisons of gene expression signatures in a principal component analysis. The adjusted cutoff threshold resulted in 81 and 29 genes with increased and decreased differential expressions, respectively. We used a heat map to compare the expression profiles of the top 50 genes, including 25 genes with the largest increased differential expression and 25 genes with the largest decreased differential expression after the 3 Hz and 1 Hz exercises (Figure 1A,C). Several genes were increased after both exercises, specifically NR4A3, EGR1, XIRP1, ABRA, NR4A1, ASB5, HSPA1B, and HSPA1A. However, no genes were commonly down-regulated from the exercise in the sessions.

A principal component analysis for the 3 Hz exercise samples resulted in a variation of 17.7% and 12.4% for the first (PC1) and second (PC2) principal components (Figure 1B). Further, there was a significant correlation between the limb (exercised versus non-exercised) and PC1 (R^2^ = 0.59, *p* < 0.001) but not between PC2 and the exercised limb (R^2^ = 0.03, *p* = 0.46). A principal component analysis for the 1 Hz exercise samples resulted in a variation of 24.8% and 10.5% for PC1 and PC2, respectively (Figure 1D). There were no correlations found between the limb (exercised versus non-exercised) and PC1 (R^2^ = 0.04, *p* = 0.44) or PC2 (R^2^ = 0.02, *p* < 0.56) after the 1 Hz electrically induced exercise.

### 2.2. Genomic Pathway Enrichment: EnrichR

The 258 and 274 increased and decreased differentially expressed genes after the 3 Hz electrically induced exercise session were submitted for pathway enrichment analysis using the EnrichR [53,54,55] algorithm to determine common pathways associated with the regulated genes. A total of 1596 pathways defined within the Reactome Knowledgebase were used for the pathway analysis, which were defined within 26 broad domains [56,57,58]. There were 87 pathways with an increased enrichment spanning 13 domains and 18 pathways with decreased enrichment spanning 7 domains after the 3 Hz electrically induced exercise. Figure 2A illustrates that Signal Transduction, Cellular Response to Stimuli, and Metabolism of Proteins were among the domains with the highest number of enriched pathways associated with increased expression. These domains included the genes EGR1, FOS, IRS2, NR4A3, VEGFA, PGC-1α, and MYC, which are strong transcriptional regulators of skeletal muscle metabolism and phenotype. Figure 2B illustrates the Metabolism of RNA domain had the highest number of enriched pathways associated with decreased expression and included genes such as LSM3, ZCRB1,and CALM1. Due to the lack of significant differentially expressed genes after the 1 Hz electrically induced exercise, an EnrichR pathway assessment was not performed.

### 2.3. Genomic Pathway Enrichment: GSEA

We performed a pathway enrichment assessment using 22,870 genes using the GSEA algorithm [33,59,60] across the Reactome Knowledge Base pathways [56,57,58]. There were 168 pathways spanning 15 domains with an increased (upregulated) enrichment after 3 Hz electrically induced exercise (Figure 3A). Similar to the EnrichR algorithm, pathways within the Signal Transduction made up most of the enriched pathways, with the associated early-response genes of PGC-1α and IRS2 contributing to the increased enrichment. Of note, within these positively enriched pathways, some genes were still found to be down-regulated. There were 57 pathways spanning 13 domains with a decreased (downregulation) enrichment after the 3 Hz electrically induced exercise (Figure 3B).

We also performed the enrichment analysis using the GSEA algorithm for the 1 Hz electrically induced exercise. There were 30 pathways spanning 10 domains with an increased enrichment (up-regulated) and 21 pathways spanning 8 domains with a decreased enrichment (down-regulated). Of note, Cellular Response to Stimuli (7 up-regulated and 0 down-regulated) and Gene Expression (6 up-regulated and 0 down-regulated) domains had the highest number of enriched pathways. This differed from the 3 Hz exercise, where the Signal Transduction domain (2 up-regulated and 4 down-regulated) had the most up-regulated pathways. Additionally, 13 up-regulated pathways were common to both the 3 Hz and 1 Hz exercises, with Cellular Responses to Stimuli being the domain with the most common pathways (6). There were 5 down-regulated enriched pathways in common between the 3 Hz and the 1 Hz exercises, with the Transport of Small Molecules and Metabolism of Proteins each having 2 common pathways. None of the common pathways were up-regulated after one exercise and down-regulated after the other.

### 2.4. A Priori Genomic Expression Signature

We selected a series of genes with key roles in human skeletal muscle pathways associated with hypertrophy (PGC1α and ABRA), atrophy (MSTN), muscle contractions (NR4A3 and ABRA), metabolism (PGC1α and NR4A3), and insulin signaling (IRS2) from the pathways that were significantly up- or down-regulated 3 h after the bout of exercise using a 3 Hz stimulation frequency. These selected genes are known to be strongly regulated after bouts of volitional and electrically induced exercise. PGC1α, a key transcriptional regulator for muscle metabolism and mitochondrial biogenesis, had a 4.86 ± 4.64-fold increase after the 3 Hz exercise with an FDR of 0.01 when comparing the expression magnitude of the exercised limb to the control limb, and a 1.76 ± 2.30-fold increase after the 1 Hz exercise with an FDR of 0.68 when comparing the expression magnitude of the exercised and control limbs (Figure 4A). NR4A3 had a 17.36 ± 17.37-fold increase after the 3 Hz exercise with an FDR of 0.0027 when comparing the expression magnitude of the exercised limb to the control limb, and a 7.33 ± 5.62-fold increase after the 1 Hz exercise with an FDR of 0.65 when comparing the expression magnitude of the exercised and control limbs (Figure 4B). ABRA had a 4.98 ± 4.82-fold increase after the 3 Hz exercise with an FDR of 0.015 when comparing the expression magnitude of the exercised limb to the control limb, and a 4.36 ± 4.77-fold increase after the 1 Hz exercise with an FDR of 0.68 when comparing the expression magnitude of the exercised and control limbs (Figure 4C). IRS2 had a 3.71 ± 2.24-fold increase after the 3 Hz exercise with an FDR of 0.0021 when comparing the expression magnitude of the exercised limb to the control limb, and a 0.99 ± 2.29-fold increase after the 1 Hz exercise with an FDR of 1.0 when comparing the expression magnitude of the exercised and control limbs (Figure 4D). MSTN had a −2.53 ± 2.22-fold decrease after the 3 Hz exercise with an FDR of 0.015 when comparing the expression magnitude of the exercised limb to the control limb, and a −0.99 ± 3.06-fold decrease after the 1 Hz exercise with an FDR of 1.0 when comparing the expression magnitude of the exercised and control limbs (Figure 4E).

### 2.5. Epigenomic Expression Signature and Methylation Age

The 616,598 methylation CpG probes were mapped onto a total of 26,644 genes. There were no differentially methylated CpG probes after either the 3 Hz or 1 Hz electrically induced exercise in the subset of participants using a FDR of 0.05, though the small sample size limits our power to discover altered CpG probes with an FDR correction. When we used a less conservative threshold for exploratory purposes (*p* ≤ 0.0001 threshold) for the 3 Hz condition, there were 7 hypo- and 8 hypermethylated sites with an increase to 151 and 121, respectively, and with a *p* ≤ 0.001. None of the genes associated with those sites were familiar in function. The hypomethylated genes were RRAS2, SELENBP1, LOC101927835, NEB, BAALC, ERV3-1, and AFF4. The hypermethylated genes were ZNF703, MYSM1, RAB3A, FAM63B, ZNF33A, PPP4R1, ASXL3, and FAM120C.

There were 105 hypomethylated and 83 hypermethylated CpG probes, respectively, after the 3 Hz electrically induced exercise, using an uncorrected *p*-value of 0.0001. Similarly, there were 15 and 22 hypo- and hypermethylated CpG probes, respectively, after the 1 Hz electrically induced exercise, using a *p*-value cutoff of 0.0001. While no CpG probes were found to be significantly hypo- or hypermethylated after our exercises, we created heat maps that depict the 25 CpG probes that were found to be hypo- and hypermethylated after 3 Hz (Figure 5A) and 1 Hz (Figure 5B) of electrically induced exercise, using a less conservative *p*-value cutoff. The primary (PC1) and secondary (PC2) principal components were 15.9% and 15.6% after 3 Hz electrically induced exercise, respectively. There was a nonsignificant correlation of PC1 (R^2^ = 0.090, *p* = 0.47) and PC2 (R^2^ = 0.40, *p* = 0.091) with the limb (exercised versus non-exercised). Similarly, the PC1 and PC2 after 1 Hz electrically induced exercise were 29.3% and 18.8% with nonsignificant correlations with the limb of R^2^ = 0.17 (*p* = 0.41) and R^2^ = 0.28 (*p* = 0.28), respectively.

In addition to differential methylation analysis, we calculated an epigenetic biological age assessment using the M.E.A.T. 2.0 algorithm. We found a correlation between the chronological age and the estimated epigenetic age for the 3 Hz (R^2^ = 0.70, *p* = 0.010) and 1 Hz (R^2^ = 0.89, *p* = 0.005) exercise sessions. There was a correlation between the number of years post injury (YPI) and estimated epigenetic age for the 3 Hz (R^2^ = 0.89, *p* = 0.005) and 1 Hz (R^2^ = 0.50, *p* = 0.12) exercise sessions. There were strong correlations between the epigenetic age and both the biological age (R^2^ = 0.72, *p* < 0.001) and YPI (R^2^ = 0.55, *p* = 0.002) after collapsing the exercise sessions (Figure 6). However, there was no difference in the estimated epigenetic age between limbs for either exercise.

Finally, we compared the methylation for an a priori selection of CpG probes that correspond to our a priori selection of key genes related to metabolism and muscle contractile pathways. The CpG probes for PGC-1α, NR4A3, ABRA, IRS2, and MSTN were cg14757717, cg04897621, cg14747011, cg08510264, and cg2452886, respectively. While no probe reached statistical significance using an FDR correction, some probes did demonstrate a subtle difference using the unadjusted *p*-value (Figure 7). After the 3 Hz electrically induced exercise, the probes for PGC-1α (*p* = 0.004; FDR = 0.26), NR4A3 (*p* = 0.007; FDR = 0.27), and ABRA (*p* = 0.14; FDR = 0.44) were slightly hypermethylated in the exercised limb compared to the control limb, while the probes for IRS2 (*p* = 0.016; FDR = 0.29) and MSTN (*p* = 0.08; FDR = 0.38) appeared to be hypomethylated. After the 1 Hz electrically induced exercise, the probes for PGC1α (*p* = 0.29; FDR = 0.51), NR4A3 (*p* = 0.41; FDR = 0.56), and ABRA (*p* = 0.083; FDR = 0.42) were slightly hypomethylated in the exercised limb compared to the control limb, while the probes for IRS2 (*p* = 0.99; FDR = 0.73) and MSTN (*p* = 0.77; FDR = 0.67) appeared unchanged. We also did not observe any significant correlations between the magnitude of gene expression and methylation level for these a priori genes in either each exercise type or when collapsed.

## 3. Discussion

We investigated the acute effect of two long-duration, low-force electrically induced exercise interventions for 1 h (3 Hz) or 3 h (1 Hz) in people living with SCI. We discovered that there was a more robust and consistent differential expression signature after the 3 Hz exercise compared to the 1 Hz exercise. Further, we determined that the differentially expressed genes after 3 Hz exercise were associated with key pathways known to control skeletal muscle phenotype. It is worth noting the increased expression of key regulatory genes, such as PGC-1a, NR4A3, EGR1, ABRA, and IRS2, and the decreased expression of MSTN after the 3 Hz exercise, which was not robustly observed after the 1 Hz exercise. Additionally, while there were no robust epigenomic changes observed after either 3 Hz or 1 Hz exercise, we did find correlations between the estimated epigenetic age and both the biological age and number of years post injury (YPI). Because participants strongly favored the 3 Hz exercise over the 1 Hz exercise for long-term training [27], we conclude that an acute bout of long-duration, low-force electrically induced exercise using a 3 Hz stimulation more effectively challenges chronically paralyzed skeletal muscle by altering the genomic expression of genes associated with improved contractile and metabolic function, and we suggest that this may be a more feasible exercise intervention as compared to the 3 Hz condition for people living with SCI.

### 3.1. Long-Duration, Low-Force Electrically Induced Exercise and Genomic Expression

An exercise bout of sufficient intensity initiates a cascade of molecular signaling pathways that are required to promote intracellular adaptions. Understanding these immediate responses offers insights into the potential long-term effects of an exercise prescription. Importantly, in this study we demonstrate that high contractile forces are not required during electrically induced exercise bouts, contrary to what has previously been demonstrated in able-bodied people [61,62,63]. This result is consistent with our previous studies that demonstrate substantial genomic alterations after a single bout of electrically induced exercise in people soon after the onset of paralysis [25,26,64]. What is unique to this study is the comparison of two different long-duration, low-force exercises. We found that the 3 Hz (1 h) dose of exercise created a stronger genomic response consistent with the greater physiological challenge as observed by the induction of muscle fatigue [27]. Additionally, this 3 Hz bout of electrically induced exercise strongly regulated a collection of pathway domains related to signal transduction, gene expression, and metabolism. These domains contain critical intra- and extracellular signaling cascades required to trigger cellular adaptions following physiologic stress.

Specifically, the selection of a priori genes are known to acutely respond to doses of exercise and are key contributors in regulating mitochondrial biogenesis and excitation–contraction coupling, among other important functions related to the metabolic and contractile phenotype of skeletal muscle [33,34,35,36,37,38,39,40,41,42,43,65,66]. Consistent with previous work, our long-duration, low-force exercise using a 3 Hz stimulation frequency resulted in a similar expression profile of these key transcriptional regulators to higher frequency protocols [26,64,67,68]. As a contrast, the electrically induced exercise with a 1 Hz stimulation frequency increased genomic expression for some individuals but remained unaltered in others and not rising to a level of significance. This suggests that the magnitude of physiologic stress from the 1 Hz exercise did not challenge paralyzed skeletal muscle to the same extent that the 3 Hz exercise did, consistent with the previously found dose response of exercise [68]. Further, MYC proto-oncogene, bHLH transcription factor (MYC), ankyrin repeat domain 1 (ANKRD1), and early growth response 1 (EGR1) were also among the genes with the largest increase in expression. Together, our collection of a priori genes and the other acute response genes highlight the effectiveness of our 3 Hz exercise prescription in causing acute changes critical for muscle development, function, and adaptions. Whether the acute changes observed after the dose of exercise described in this study are suitable to transition muscle phenotype from fast glycolytic to slow oxidative remains unknown and a focus of future investigations. However, these results support that this type of exercise creates an acute gene expression signature consistent with responses that, when done chronically, cause changes in phenotype.

### 3.2. Long-Duration, Low-Force Electrically Induced Exercise and Epigenomics

We did not detect any robust methylation changes using a conservative threshold after either long-duration exercise. Previous findings have demonstrated hypomethylation across the epigenome, including of PGC-1α, after bouts of exercise [69,70,71,72,73].

### 3.3. Clinical Considerations and Limitations of This Study

A primary goal of this study was to develop an effective and safe bout of exercise for people living with SCI. Critical to the development of any exercise prescription is compliance. Therefore, we previously surveyed participants to determine the form of exercise that would likely lead to the highest level of compliance. The 3 Hz exercise was found to be well-tolerated and was preferred by our participants compared to the 1 Hz exercise session. Another vital consideration for exercise prescription is feasibility [27]. We must be cognizant of the barriers, which can often be insurmountable, that prevent people living with SCI from participating in safe, effective bouts of routine exercise. People in rural communities and in lower socioeconomic levels cannot readily access expensive specialty exercise equipment tailored for people with SCI. We designed our low-force exercise to address these obstacles. Specifically, we used commercially available muscle-stimulation units, allowing the exercise to be performed at home. Second, because people living with SCI have undergone substantial deterioration of the skeletal system, our exercise limits muscle force production even when performed in the seated position where shear force is maximized [74]. Minimizing force production while adequately challenging muscle activity is critical to ensuring a safe and efficacious exercise prescription for people with chronic SCI. Finally, our 3 Hz exercise adequately stresses skeletal muscle, as supported by the robust changes in muscle fatigue observed previously [27] and as supported by the genomic changes demonstrated in this study.

There is an immediate clinical need to implement exercise prescriptions for people living with SCI. However, caution is required to ensure that exercises are safe and efficacious. Our study suggests that a long-duration, low-force electrically induced exercise using a 3 Hz stimulation frequency is most effective in altering acute gene expression signatures. However, we must consider the limitations of this study as we move to implement an intervention for clinical use. First, we elected to control the dose of exercise by delivering the same number of stimulus pulses to the muscle during each exercise session and modulating the rate at which those stimulus pulses were delivered. Therefore, the duration of the exercise session was three times longer during the 1 Hz exercise session compared to the 3 Hz session. Second, we performed the muscle biopsies 3 h after the completion of the exercise session. Therefore, the biopsy after the 1 Hz exercise session was nearly 6 h after the start of the exercise session compared to 4 h for the 3 Hz exercise session. Previous work from our lab and others have demonstrated that the 3 h post-exercise time window captures expression changes for early-response genes such as PGC1a and NR4A3. However, because the 1 Hz exercise session lasted 3 h, we may have missed the peak expression changes of these early-response genes, leading to the muted expression profile observed in the 1 Hz condition. Because significant fatigue occurred after only 1000 pulses in both the 3 Hz (75% fatigued) and 1 Hz protocols (40% fatigue) [27], the biopsy was taken nearly 5 h after the major loss of force that occurred in the 1 Hz condition, whereas the biopsy was taken ~3.5 h after the major loss of force that occurred in the 3 Hz condition. This may explain the attenuated gene expression under the 1 Hz condition and warrants further study. Third, we did not observe any robust and consistent change in DNA methylation after either form of exercise. We were unable to perform the DNA methylation analysis on all participant samples because we prioritized tissue samples for RNA expression analysis and there was too little sample of sufficient quality remaining for some participants. Therefore, our acute methylation changes were based on a limited sample and support a further study with a larger sample size to increase power. When we lowered our thresholds, we did find significant DNA methylation, but among genes with no known impact on skeletal muscle function. Finally, DNA methylation occurs on a different time constant compared to gene expression changes. Because multiple biopsies afford some risk to skin integrity among people with paralysis, we prioritized gene expression analysis at just the 3 h time window but may have missed the optimal time to optimize the signal–noise ratios. Muscle samples analyzed immediately after a bout of exercise have been shown to offer more robust directional shifts in methylation levels [64,75].

## 4. Methods

### 4.1. Participants

Ten people with a complete SCI (ASIA-A) participated in this study (Table 1). Participants were not fasting and consumed their last meal one hour before the start of the study. All but one participant completed both the 1 h and 3 h unilateral electrically induced exercise session using a 3 Hz and 1 Hz stimulation frequency, respectively. All participants refrained from using any form of electrical muscle stimulation for at least 1 year prior to study enrollment. Further, individuals were excluded from participation if they had a recent history of bone fracture, pressure ulceration, recent urinary tract infection, or other musculoskeletal or integument lesions. All participants provided written informed consent approved by the University of Iowa institutional review board (IRB# 201503732) in accordance with the Declaration of Helsinki. Clinical Trial: NCT03139344.

### 4.2. Electrically Induced Exercise Protocol

Participants completed bouts of unilateral electrically induced exercise of the quadriceps muscle using a 3 Hz or 1 Hz stimulation frequency while the opposite limb remained as a non-exercised control (Figure 8). One limb was randomly selected to perform the unilateral electrically induced exercise using either the 3 Hz or 1 Hz stimulation frequency, while the opposite limb served as the non-exercised control limb. In the subsequent session, the non-exercised leg from the first session was exercised using the remaining stimulation frequency (3 Hz or 1 Hz), while the opposite limb served as the non-exercised control limb. Exercise sessions were separated by at least 6 weeks to allow for a complete wash out from the bout of exercise.

Each exercise session consisted of six trains of 2000 stimulus pulses delivered at either 3 Hz or 1 Hz. A 1 min recovery period was provided between each train. Stimulus pulses were delivered using a constant current muscle stimulator (Digitimer model, DS7A, Digitimer, Welwyn Garden City, Hertfordshire, UK) controlled by a computer, with participants positioned in a wheelchair with hips, knees, and ankles placed at approximately 90° of flexion and restrained to minimize limb movement. Supramaximal stimulus intensities were used during each exercise session, ensuring that nearly all muscle fibers were recruited during each electrically induced muscle contraction. The electrode placement was carefully positioned in relation to the lateral epicondyle and greater trochanter and marked with an indelible marker. The electrically induced exercise session using the 3 Hz stimulation frequency lasted approximately 1 h, while the electrically induced exercise session using a 1 Hz stimulation frequency lasted approximately 3 h. The bilateral percutaneous muscle biopsies were performed 3 h after the conclusion of the exercise. Therefore, the muscle biopsy was performed at approximately 4 h and 6 h after the start of the exercise session for the 3 Hz and 1 Hz exercises, respectively. The dose of 12,000 stimulus pulses was selected from our previous work that demonstrated robust changes in gene expression after higher-frequency electrically induced exercise in human paralyzed muscle [25,26,48].

### 4.3. Muscle Biopsy Procedure

Three hours after the completion of either exercise session, a percutaneous muscle biopsy was performed using our previously reported protocol [25,48]. In brief, muscle samples were taken from the exercised and non-exercised limb using a Temno biopsy needle (T1420, Cardinal Health, Inc., Dublin, OH, USA) under ultrasound guidance within a sterile field. Four to five passes of the needle were made to increase the sampling range within the muscle. Each pass of the needle was made through the same cutaneous incision site, but the needle angle was altered to sample different parts of the vastus lateralis muscle. Each pass of the needle yielded between 10 and 20 mg of muscle tissue. Following harvest, muscle samples were placed in RNALater (Ambion^®^, ThermoFisher Scientific Inc., Waltham, MA, USA) and stored at −80 °C until RNA or DNA extraction.

### 4.4. RNA Extraction and Gene Expression Analysis

RNA extraction was performed using the RNEasy Fibrous Tissue Kit (Qiagen, Hilden, Germany) with DNAse to remove genomic DNA from final samples according to the manufacturer’s protocol, as we have previously reported [25,48,64]. Microarray hybridizations were performed using Human Transcriptome 2.0 (HTA) microarrays (Affymetrix, Inc., Santa Clara, CA, USA; Part No. 902233) according to the protocol supplied by the manufacturer. Arrays were scanned with the Affymetrix Model 3000 scanner with 7G upgrade, and data were collected using the GeneChip operating software (GCOS) v1.4.

HTA hybridization data were imported, normalized, mean-summarized with a Robust Multi-array Average using Bioconductor (v3.9), implemented through the R programming language (v3.5.1), and converted into a log2 signal intensity for each gene transcript [76,77]. Subsequently, differential gene expression was evaluated using three methods. First, we performed a principal component analysis across all samples and all gene transcripts for each type of exercise: 3 Hz electrically and 1 Hz electrically induced exercises. We calculated a correlation coefficient of the major principal components across several factors, including limb (exercise limb versus non-exercised limb), lesion level (paraplegic versus tetraplegic), age, and years post injury. Next, we used two pathway enrichment analyses, (1) the Enrichr algorithm [53,54,55] and (2) the gene set enrichment algorithm (GSEA) [33,59,60], to develop an expression signature after the 3 Hz or 1 Hz electrically induced exercise session. A total of 1596 pathways defined across 26 biological domains defined within the Reactome Pathways Knowledgebase were used for both pathway enrichment analyses [56,57,58]. Pathways associated with the Disease and Drug domains were removed from the analysis.

For the Enrichr pathway enrichment analysis, we submitted the set of up- and down-regulated genes to the Enrichr server [53,54,55] to compute the overlap with the annotated genesets defined within Reactome Pathways Knowledgebase [56,57,58]. The computed enrichment statistics were as follows: (1) Fisher Exact Test, (2) a custom z-score computed from the deviation of the expected rank by the Fisher exact test, and (3) a combined score computed from method 1 and 2; these were utilized to determine the enriched pathways [53,54,55].

Differential expression was determined by a paired sample *t*-test of the exercised versus non-exercised limbs, which was performed within each exercise session with the resulting *p*-values adjusted by multiple testing correction using the false discovery rate (FDR) procedure. Genes with a FDR below 0.05 and a fold-change above 1.25 indicated an increased differential expression after exercise; those with a fold-change below 0.8 (1/1.25) indicated a decreased differential expression after exercise. An enrichment statistic was calculated using the Enrichr algorithm with a threshold of 0.05 to determine enriched pathways.

Log2 hybridization values for all genes were imported into the GSEA application (version 4.3.2) to compare pathway signatures between the exercised and non-exercised limb within the 3 Hz and 1 Hz electrically induced exercise session, respectively [33,59,60]. GSEA was performed on all defined pathways regardless of the number of genes defined within the pathway. Additionally, permutations were across phenotypes (exercised and non-exercised limb), with a permutation number set to 1000.

Finally, the differential expressions of a selection of a priori genes (NR4A3, PGC-1α, ABRA, and MSTN) that are known to be responsive after acute bouts of exercise were compared for each electrically induced exercise session [25]. These were selected because of their role in the regulation of skeletal muscles’ metabolic and contractile phenotypes. A one-factor studentized *t*-test was used to determine differences across exercise sessions.

### 4.5. DNA Extraction and Methylation Expression Analysis

DNA methylation analysis was performed on a subset of seven people who had a sufficient sample quantity and quality for analysis after 3 Hz and 1 Hz exercise, respectively. DNA was extracted from a subset of samples (exercise and control) using the DNEasy Tissue Kit (Qiagen, Hilden, Germany) following the manufacturer instructions, as we have previously reported [64]. Extracted DNA was treated with sodium bisulphite using the EZ DNA methylation kit (Zymo Research, Irvine, CA, USA). DNA methylation was quantified using the Infinium Human MethylationEPIC BeadChip (Illumina, Inc., San Diego, CA, USA) using the manufacturer recommended protocol.

Raw IDAT files were processed using Bioconductor and the missMethyl package in the R statistical programming environment [76,77,78,79]. After normalization of probes using a subset quantile algorithm, CpG probes related to single-nucleotide polymorphisms or with a high cross-reactive or non-specific probe were removed from further analysis. The remaining probes were converted into a β-value representing the ratio of the methylated probe intensity to the overall intensity of both the methylated and unmethylated probe intensities and used for biological interpretation. β-values were converted into M-values for all statistical testing procedures.

A principal component analysis was performed within each type of electrically induced exercise (3 Hz and 1 Hz) and correlated to factors that might explain the variance between samples. A biological age was calculated for each limb using the methylation epigenetic age test version 2.0 (MEAT 2.0) implemented through Bioconductor [80,81], and differences were compared between the exercised and control (unexercised) limbs.

A paired-sample *t*-test was performed within each type of electrically induced exercise, and the resulting *p*-value was adjusted using the FDR procedure for multiple testing comparisons. CpG probes with an FDR below 0.05 and at least a 20% change in methylation between the exercised and non-exercised limbs were mapped onto the corresponding gene. Lastly, we evaluated differences in methylation for the a priori genes (PGC-1α, NR4A3, ABRA, IRS2, and MSTN) and correlated changes in methylation to changes in gene expression. The median methylation value was calculated from all CpG probes corresponding to these genes.

### 4.6. Statistical Analysis

GSEA and EnrichR use an enrichment statistic and false detection rate (FDR) to define pathways that were significantly altered between the exercised and non-exercised limbs for both genomic and epigenomic expression changes. We performed a paired *t*-test in a set of a priori genes to compare the gene expression fold-change or gene methylation difference between the 3 Hz and 1 Hz exercises. Lastly, we calculated a correlation coefficient between the genomic and epigenomic change for a selection of a priori genes. A significance threshold of 0.05 was used for all testing procedures, and all results are reported as a mean ± standard deviation unless otherwise noted.

## 5. Conclusions

Exercise is a fundamental physiologic stress for promoting healthy skeletal muscle and may be underutilized in people living with SCI. The goal of this study was to determine the acute genomic and epigenomic changes following two low-force and long-duration electrically induced exercises suitable for people living with chronic SCI to perform at home. We found that electrically induced exercise utilizing a 3 Hz frequency adequately stressed human paralyzed skeletal muscle and demonstrated a robust and significant change in gene expression consistent with other forms of volitional exercise in people without SCI. These results support the use of this long-duration exercise for people living with SCI, though additional work is needed to determine if training with this form of exercise changes muscle phenotype and reduces health risks in people living with SCI.

## Figures and Tables

**Figure 1 ijms-25-10189-f001:**
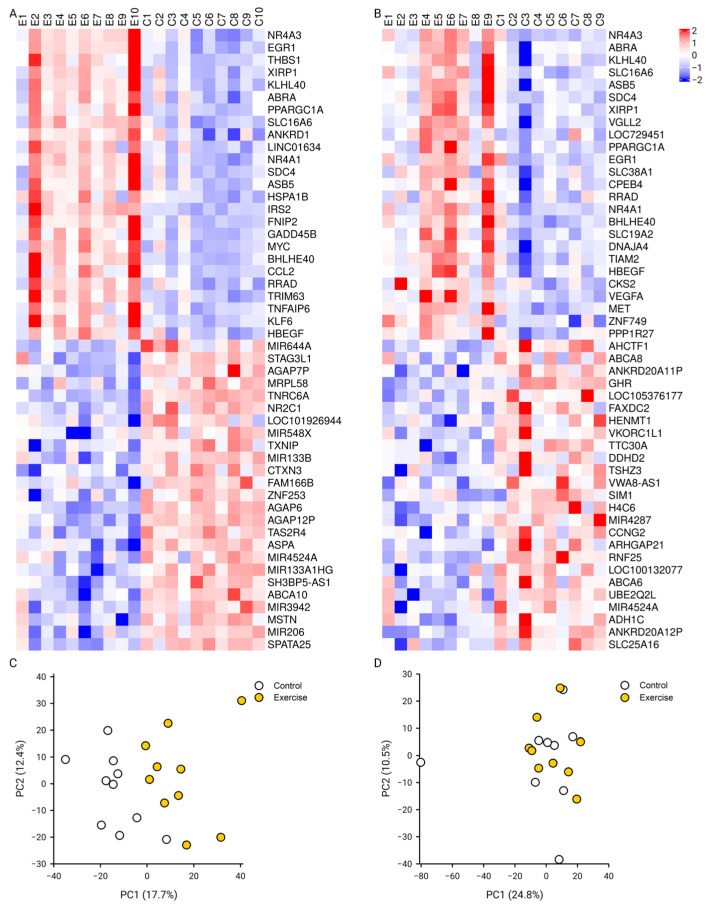
Gene expression maps. The 25 most increased (red) and 25 most decreased (blue) genes are represented. (**A**) Heat map of the 50 genes with the largest change in expression (FDR < 0.05) 3 h after the end of the long-duration, low-force exercise with a 3 Hz stimulation frequency in people with SCI. (**B**) Heat map of the 50 genes with the largest change in expression (*p* < 0.004) 3 h after the end of the long-duration, low-force exercise with a 1 Hz stimulation frequency in people with SCI. Principal component biplots showing the grouping between the exercised and non-exercised control limbs for the first and second principal components of all genes 3 h after the long-duration, low-force exercise with a 3 Hz stimulation frequency (**C**) and a 1 Hz stimulation frequency (**D**).

**Figure 2 ijms-25-10189-f002:**
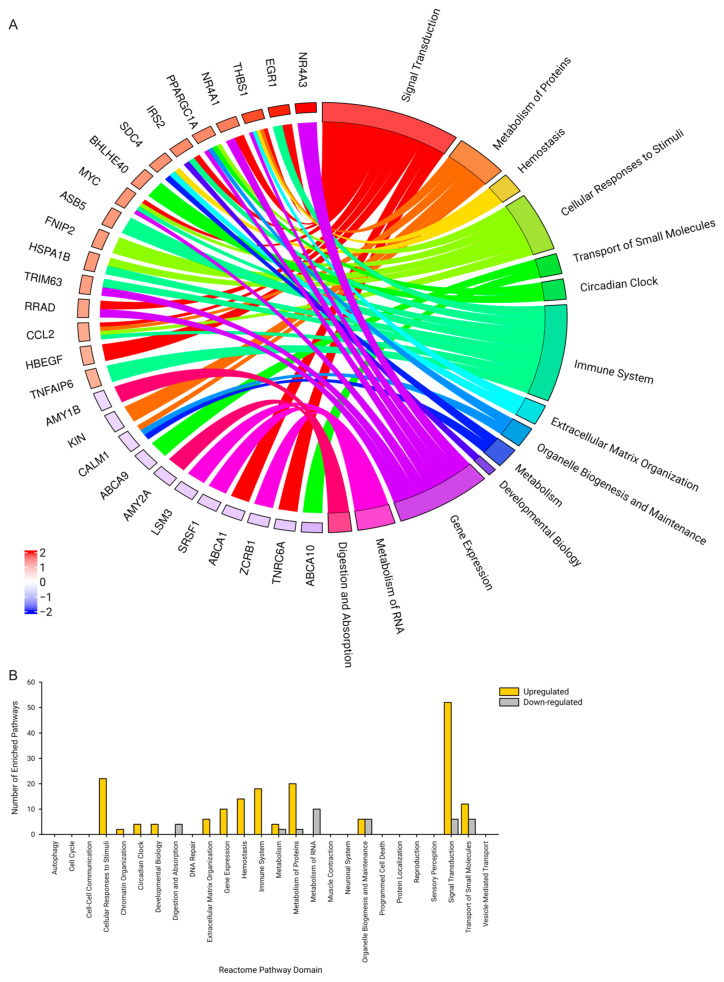
EnrichR expression signature after 3 Hz electrically induced exercise. A total of 87 Reactome pathways across 13 domains were found to be positively enriched using the EnrichR algorithm after 3 Hz electrically induced exercise. A total of 18 Reactome pathways across 7 domains were found to be negatively enriched using the GSEA algorithm after 3 Hz electrically induced exercise. A cutoff of 0.05 for the calculated *p*-value was used. (**A**) A chord plot showing the relationship between the enriched Reactome domains and the genes that were up-regulated (red) or down-regulated (blue) 3 h after the 3 Hz exercise. (**B**) A bar plot showing the number of pathways enriched within each domain pathway.

**Figure 3 ijms-25-10189-f003:**
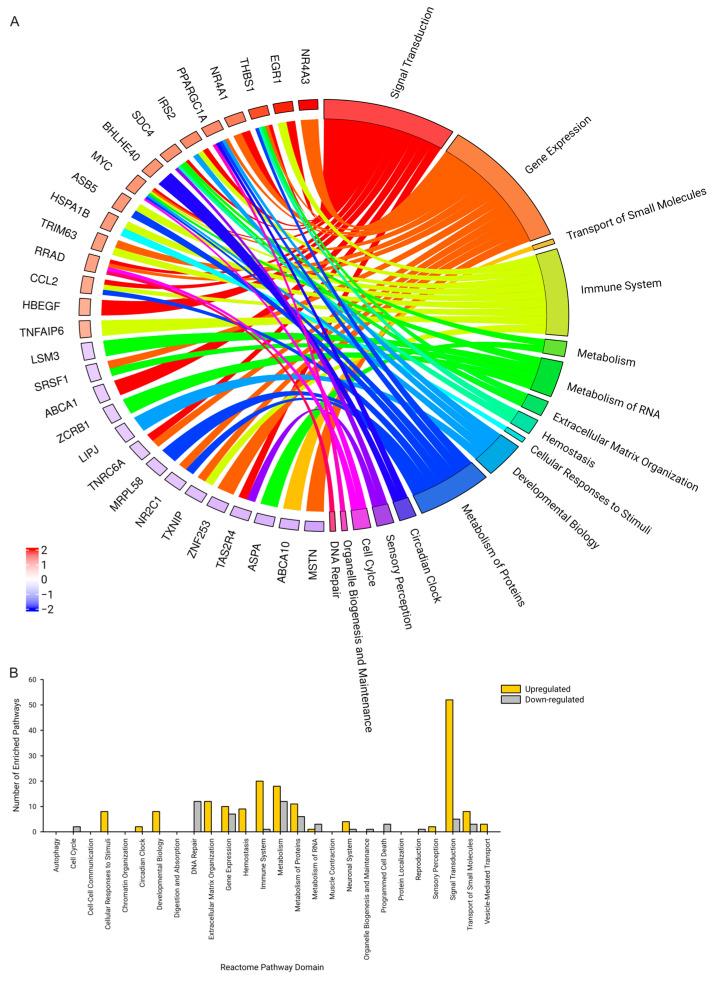
GSEA expression signature after 3 Hz electrically induced exercise. A total of 168 Reactome pathways across 15 domains were found to be positively enriched using the GSEA algorithm after 3 Hz electrically induced exercise. A total of 57 Reactome pathways across 13 domains were found to be negatively enriched using the GSEA algorithm after 3 Hz electrically induced exercise. A cutoff of 0.05 for the calculated *p*-value was used. (**A**) A chord plot showing the relationship between the enriched Reactome domains and the genes that were up-regulated (red) or down-regulated (blue) 3 h after the 3 Hz exercise. (**B**) A bar plot showing the number of pathways enriched within each domain pathway.

**Figure 4 ijms-25-10189-f004:**
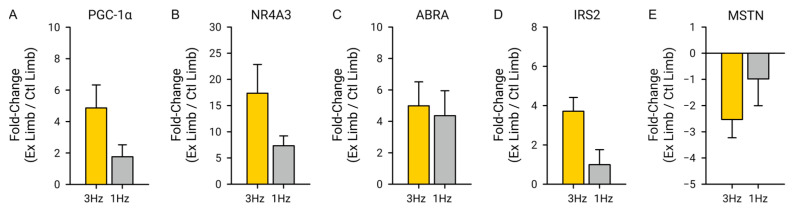
A priori gene expression fold-change. (**A**) There was a significant increase in the expression of PGC1α after the 3 Hz exercise (FDR = 0.01) relative to the control (unexercised) limb, but there was not a significant increase of PGC1α after the 1 Hz exercise (FDR = 0.68) relative to the control (unexercised) limb. (**B**) There was a significant increase in the expression of NR4A3 after the 3 Hz exercise (FDR = 0.0027) relative to the control (unexercised) limb, but there was not a significant increase of NR4A3 after the 1 Hz exercise (FDR = 0.65) relative to the control (unexercised) limb. (**C**) There was a significant increase in the expression of ABRA after the 3 Hz exercise (FDR = 0.015) relative to the control (unexercised) limb, but there was not a significant increase of ABRA after the 1 Hz exercise (FDR = 0.68) relative to the control (unexercised) limb. (**D**) There was a significant increase in the expression of IRS2 after the 3 Hz exercise (FDR = 0.0021) relative to the control (unexercised) limb, but there was not a significant increase of IRS2 after the 1 Hz exercise (FDR = 1.0) relative to the control (unexercised) limb. (**E**) There was a significant decrease in the expression of MSTN after the 3 Hz exercise (FDR = 0.015) relative to the control (unexercised) limb, but there was not a significant decrease of MSTN after the 1 Hz exercise (FDR = 1.0) relative to the control (unexercised) limb.

**Figure 5 ijms-25-10189-f005:**
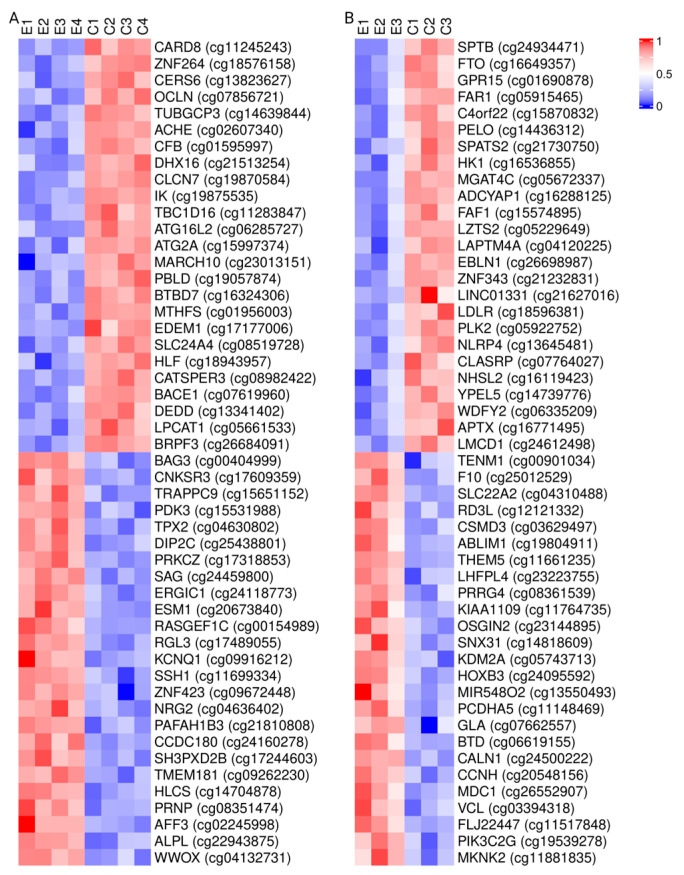
Heat map of post-exercise DNA methylation. A heat map of the 25 most hypermethylated (red) and 25 most hypomethylated (blue) CpG probes and corresponding genes 3 h after the end of a long-duration, low-force 3 Hz electrically induced exercise (**A**) and 1 Hz electrically induced exercise (**B**) for people with a spinal cord injury. While probes reached a significant differential level using a FDR cutoff of 0.05, these probes did have a differential methylation below an uncorrected *p*-value of 0.001 between the exercised and non-exercised (control) limbs.

**Figure 6 ijms-25-10189-f006:**
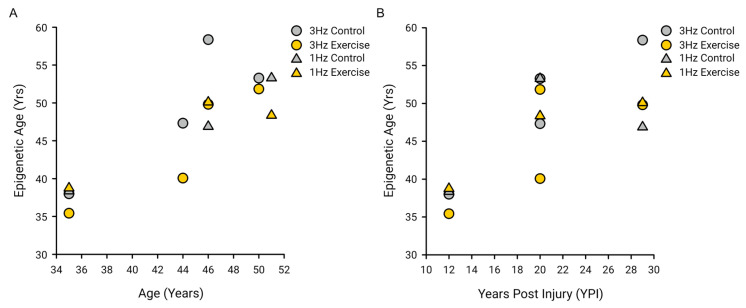
Epigenetic age correlations. (**A**) There were strong correlations between the chronological age and estimated epigenetic age for the 3 Hz (R^2^ = 0.70, *p* = 0.010) and the 1 Hz (R^2^ = 0.89, *p* = 0.005) exercise sessions as well as when the exercise sessions were collapsed (R^2^ = 0.72, *p* < 0.001). (**B**) There were strong correlations between the number of years post injury (YPI) and estimated epigenetic age for the 3 Hz (R^2^ = 0.89, *p* = 0.005) and the 1 Hz (R^2^ = 0.50, *p* = 0.12) exercise sessions as well as when the exercise sessions were collapsed (R^2^ = 0.55, *p* = 0.002).

**Figure 7 ijms-25-10189-f007:**
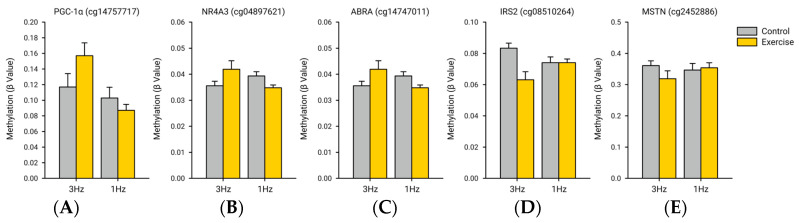
Relative methylation levels for a selection of a priori CpG probes. While no probe reached statistical significance using an FDR correction, some probes did demonstrate a subtle difference using the unadjusted *p*-value. After the 3 Hz electrically induced exercise, the probes for (**A**) PGC-1α (cg14757717; *p* = 0.004; FDR = 0.26), (**B**) NR4A3 (cg04897621; *p* = 0.007; FDR = 0.27), and (**C**) ABRA (cg14747011; *p* = 0.14; FDR = 0.44) were slightly hypermethylated in the exercised limb compared to the control limb, while the probes for (**D**) IRS2 (cg08510264; *p* = 0.016; FDR = 0.29) and (**E**)MSTN (cg2452886; *p* = 0.08; FDR = 0.38) appeared to be hypomethylated in the exercised limb relative to the control limb. After the 1 Hz electrically induced exercise, none of the a priori probes were found to differ between the exercised and control limbs.

**Figure 8 ijms-25-10189-f008:**
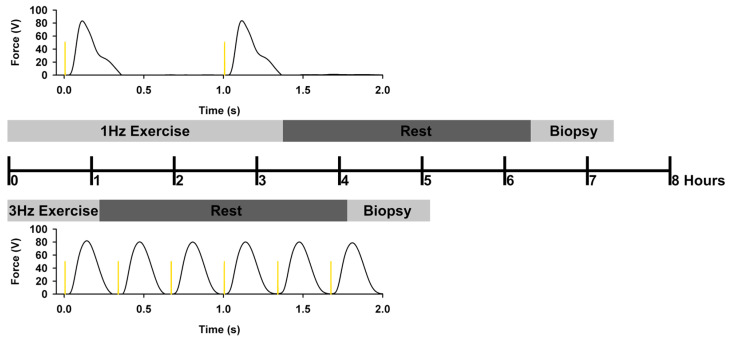
Study design and example of force response from our previous report [27]. Note that for every stimulus pulse delivered during the 1 Hz exercise, three stimulus pulses were delivered during the 3 Hz exercise. The number of stimulus pulses delivered to the muscle was kept constant for the 1 Hz and 3 Hz exercise sessions; therefore, the time required to complete the 1 Hz exercise session (~3 h) was three times longer than the time required to complete the 3 Hz exercise session (~1 h).

**Table 1 ijms-25-10189-t001:** Participant demographics and anthropometrics.

Exercise Protocol	Participant Number	Injury Level	Age (Yrs)	Years Post Injury (Yrs)	Height (cm)	Mass (kg)
1 Hz	9 Male	3 Paraplegic 6 Tetraplegic	39.5 ± 8.0	13.1 ± 9.1	183.4 ± 3.1	85.4 ± 21.1
3 Hz	10 Male	4 Paraplegic 6 Tetraplegic	38.8 ± 7.8	13.0 ± 8.6	183.9 ± 3.3	85.3 ± 20.0

## Data Availability

Data presented herein can be made available by contacting the corresponding author. Raw genomic and epigenomic files have been deposited in NCBI gene expression omnibus (Accession Number Pending).

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
