# Peer review of "Distinct Genomic Expression Signatures after Low-Force Electrically Induced Exercises in Persons with Spinal Cord Injury"

_ijms, 2024, doi:10.3390/ijms251810189_

Round 1

Reviewer 1 Report

Comments and Suggestions for Authors

Petrie et al (IJMS-3161584) use two models of electrical stimulation in individuals with motor and sensory complete spinal cord injury (AIS A) to test the magnitude of transcriptomic (via array) and methylomics (via array) in paralyzed muscle. Their data suggests when controlled for number of stimuli (6 trains of 2,000 pulses), a shorter duration protocol with quicker (3 Hz; ~1 h) provides greater molecular changes compared to longer duration with slower stimuli (1 Hz; ~3 h). I hope they find the following helpful.

Strengths

-Cross-over design with ‘work’ normalized to number of stimuli/contractions is a strong approach

-Use of AIS A participants improves rigor

-Addition of important high resolution molecular work in a scientific area greatly lacking such info, especially in regards to muscle

Major limitations

-It is unclear if this is a new clinical study by the investigators or if they are leveraging biospecimens from a previous study. They cite the feasibility study of using 1 Hz or 3 Hz in a shorter duration study, but there’s no table describing the participants (sex, age, weight, level/duration of injury, etc) that would be expected. I wasn’t able to find any citation within the manuscript that may link this info.

-If this is a new study, changes in max/mean/final force for each train and the overall means for each are an important consideration to show the magnitude of fatigue reached by participants. I’m assuming these are isometric contractions but clearly describing would be appreciated.

-Not having aged-matched able-bodied controls is a major limitation.

-I certainly don’t question the data, but it is surprising that upwards of 3.5 h of consistent contractions did not result in any changes in transcriptomics in chronically paralyzed muscle, even if it is low force. This group’s previous data show some fatigue after 1000 pulses at 1 hz (fatigue index of ~.65) that would be expected to be result in even more fatigue over 12000 pulses. The 3 h timepoint post-contraction should be a prime window for changes in gene expression. Log transformed array intensity and t-tests vs sequencing read counts and negative binomial comparisons may result in some loss of resolution that could explain this. An alternative could be using the R package limma downstream of RMA and utilizing the general linear model within the package as it was originally created for analysis of array based gene expression assays. It is understood this is not an easy lift if the investigators are not familiar with this package, but it does provide an alternative.

Moderate limitations

-The description of Enrichr can be clarified. As far as I understand, Enrichr itself does not have a gene set enrichment algorithm or its own way to adjust p values. It uses computationally efficient process and an intuitive GUI to run and display the results of other well-known annotated gene sets. The methods the authors describe is clear…DEGs were put into Enrichr and Reactome was used. But to me, the data are analyzed using Reactome’s algorithms, not Enrichr’s. I admittedly could be misunderstanding how Enrichr processes statistics as it goes through all the different enrichment and pathway sets (i.e. Enrichr correcting its p values across all pathways so that statistics from the Reactome data accessed through Enrichr is different than using Reactome on its own.

-The exploratory methylomics is appreciated but majorly underpowered and would not be expected to yield usable data with the very conservative thresholds used in this manuscript. I would suggest the authors consider approaching the methylomics as explicitly exploratory and expand FDR or use a nominal p value cutoff and reduce percent methylation change. 

Minor limitations

-Was this trial registered in ClinicalTrials.gov? If so, please provide identified. If not, please state so explicitly. 

-Please provide IRB approval number

-There is no explicit mentioning of whether participants arrived fasted or remained fasted for the duration of the on-site portion of the study. While it may seem inconsequential, participants were on-site for 5-8 h. If they arrived after an overnight fast or something, it’s reasonable gene expression could in part be altered by 16-18 h fast. 

- It is okay the investigators expanded the statistical threshold for exploratory analyses of the 1 hz group but how they ended up with p < 0.004 is unclear.

-No GEO upload or supplemental intensity matrix are available.

-Some incomplete sentences, some cut/copy/paste errors, typos. Simple stuff corrected on a strong proofread.

Reviewer 2 Report

Comments and Suggestions for Authors

This manuscript has certain clinical significance for the treatment of SCI. The author should explain in the discussion of the manuscript how the experimental conclusions were confirmed without conducting validation experiments? Also, it is suggested that the author delete Fig5 and mention the results in the discussion. Additionally, P3, L121, and Fig1B should be Fig1C.

Comments on the Quality of English Language

Minor editing of English language required.

Round 2

Reviewer 1 Report

Comments and Suggestions for Authors

I thank the authors for strongly considering the reviewer suggestions and taking the time and effort to thoroughly address them. I have no further comments.